The relationship between carbon and nitrogen metabolism in cucumber leaves acclimated to salt stress

Naliwajski Marcin Robert marcin.naliwajski@biol.uni.lodz.pl
Skłodowska Maria
Department of Plant Physiology and Biochemistry, Faculty of Biology and Environmental Protection, University of Lodz , Lodz , Poland
Anderson Todd
Electronic publication date: 2018 Dec 10
Publication date: 2018
Volume: 6
Electronic Location ID: e6043
Received 2018 May 16; Accepted 2018 Oct 30
Copyright: ©2018 Naliwajski and Skłodowska
Copyright year: 2018
Copyright holder: Naliwajski and Skłodowska
License: This is an open access article distributed under the terms of the Creative Commons Attribution License, which permits unrestricted use, distribution, reproduction and adaptation in any medium and for any purpose provided that it is properly attributed. For attribution, the original author(s), title, publication source (PeerJ) and either DOI or URL of the article must be cited.
License URL: https://creativecommons.org/licenses/by/4.0/

Keywords: Acclimation, Cucumber, Salt stress, Carbon metabolism, Nitrogen metabolism

Funding: University of Lodz B1711000000052.01 Polish Ministry of Science and Higher Education NN 302117735 This work was supported by University of Lodz Grant No B1711000000052.01 and by the Polish Ministry of Science and Higher Education Grant No NN 302117735. The funders had no role in study design, data collection and analysis, decision to publish, or preparation of the manuscript.

==============================
The study examines the effect of acclimation on carbon and nitrogen metabolism in cucumber leaves subjected to moderate and severe NaCl stress. The levels of glucose, sucrose, NADH/NAD+-GDH, AspAT, AlaAT, NADP+-ICDH, G6PDH and 6GPDH activity were determined after 24 and 72 hour periods of salt stress in acclimated and non-acclimated plants. Although both groups of plants showed high Glc and Suc accumulation, they differed with regard to the range and time of accumulation. Acclimation to salinity decreased the activities of NADP+-ICDH and deaminating NAD+-GDH compared to controls; however, these enzymes, together with the other examined parameters, showed elevated values in the stressed plants. The acclimated plants showed higher G6PDH activity than the non-acclimated plants, whereas both groups demonstrated similar 6PGDH activity. The high activities of NADH-GDH, AlaAT and AspAT observed in the examined plants could be attributed to a high demand for glutamate. The observed changes may be required for the maintenance of correct TCA cycle activity, and acclimation appeared to positively influence these adaptive processes.

Introduction

Soil salinization is an environmental stress factor that can limit growth, development and productivity of plants (Bartels & Dinakar, 2013; Roychoudhury, Banerjee & Lahiri, 2015). A range of strategies have been proposed for the development of NaCl-tolerant plants. One potential tool is based on the in vitro development and isolation of NaCl-tolerant cell/callus lines by Agrobacterium-mediated transformation. Alternatively, NaCl seed priming and/or seedling conditioning can be used to increase the capacity of plants to adapt to salinity. Such low NaCl concentration pre-treatments have been shown to improve seed germination, seedling emergence and plant growth under saline conditions for a range of crop plants. The favorable effects associated with NaCl pre-treatments have been observed to persist in later development stages such as fruiting and seed production in many plants (Hossain et al., 2007; Sivritepe et al., 2008).

In plants, NaCl stress has a strong influence on nitrogen and carbon metabolism, and this is reflected in a number of changes occurring in a range of physiological and biochemical processes (Ashraf & Harris, 2004; Anjum et al., 2017). Plants can cope with salt stress by synthesizing such osmolytes as proline (Pro), soluble sugars and amines, and the osmoprotectant activities of these compounds in have been well examined in higher plants (Muchate et al., 2016; Negrão, Schmöckel & Tester, 2017). Sugars, especially glucose (Glc) and sucrose (Suc), play a crucial role in plant metabolism. They supply carbon and energy, but also participate in the signaling pathway initiating the up-regulation of defence-related genes and down-regulation of photosynthetic gene expression (Gibson, 2000; Mhamdi et al., 2010; Leterrier et al., 2012; Leterrier et al., 2016). In a wide range of plants, total carbohydrate content increases after NaCl treatment, mainly due to elevated Suc and Glc levels (Singh et al., 2015).

Under salinity stress, Suc and Glc accumulation allows active osmotic adjustment, thus facilitating adaptation by sodium translocation and compartmentation, and by influencing protein turnover and compatible solute production (Singh et al., 2015). There is strong evidence that glucose and fructose play a role during the relief period (Liu & Van Staden, 2001).

High NaCl levels can also influence plant metabolism by interfering with steps in nitrogen assimilation, thus reducing the nitrogen level in the plant (Flores et al., 2004; Debouba et al., 2006; Debouba et al., 2007). Apart from the legume family, most higher plants fix nitrogen by first reducing its inorganic form (NO3−) to NH4+ and then incorporating it into an organic form (Oliveira et al., 2009; Hachiya & Sakakibara, 2017). This process takes place via two pathways, the first is the glutamine synthetase/glutamate synthase (GS/GOGAT) cycle, yielding glutamine and glutamate (Glu) (Miflin & Habash, 2002; Liu & Von Wirén, 2017), and the other is the glutamate dehydrogenase (GDH, EC 1.4.1.2) pathway; this catalyzes the amination of 2-oxoglutarate (2-OxG) or reversible deamination of glutamate, and requires NADH or NAD+ coenzymes, respectively (Wang et al., 2014). GDH is mainly located in mitochondria but some evidence suggests that it is also present in the cytosol (Terce-Laforgue et al., 2004; Tercé-Laforgue et al., 2013; Fontaine et al., 2006). NADH-GDH plays a crucial role in ammonium detoxification because GDH displays greater aminating activity at higher levels of NH4+ (Frechilla et al., 2002; Wang et al., 2014). NAD-GDH has been proposed to play a role in the deamination of glutamate under stress and during senescence (Bechtold, Pahlich & Lea, 1998).

The tricarboxylic acid (TCA) cycle is the main source of carbon skeletons required for NH4+ assimilation, and is linked to amino acid metabolism by glutamate dehydrogenase (Hodges, 2002; Tercé-Laforgue et al., 2013). The TCA cycle thus links carbon and nitrogen metabolisms: organic acids originating from glycolysis are oxidised and directly exported as 2-OxG, which are required for NH4+ assimilation and for forming the main carbon skeletons for amino acid synthesis (Nunes-Nesi et al., 2013). NAD-GDH triggers the oxidation of Glu to 2-OxG, which may enter the TCA cycle; these play a significant role in the delivery of carbon skeletons when their amount is limited (Nunes-Nesi et al., 2013; Liu & Von Wirén, 2017). 2-OxG, an important keto-acid of the TCA cycle, plays a central role in amino acid formation and nitrogen transport (Zhang et al., 2009).

Isocitrate dehydrogenase (ICDH) and aspartate aminotransferase (AspAT, EC 2.6.1.1) are both sources of 2-OxG (Reda, 2015). While the ICDH pathway allows for net glutamate synthesis via the GS/GOGAT cycle, the AspAT pathway results in the synthesis of aspartate instead of glutamate, and requires oxaloacetate as carbon-skeleton input (Gálvez, Lancien & Hodges, 1999; Forde & Lea, 2007). Eukaryotic cells possess two forms of isocitrate dehydrogenase with different cofactor specificity: i.e.,  NAD+ (NAD+-ICDH, EC 1.1.1.41) or NADP+ (NADP+-ICDH, EC 1.1.1.42). Both forms of the enzyme catalyze the reversible oxidative decarboxylation of isocitrate to form 2-OxG. In plants, the NAD+-ICDH isoform is a TCA cycle enzyme located in the mitochondrial matrix (Douce & Neuburger, 1989) whereas NADP+-ICDH is located in the cytosol, mitochondria, plastids and peroxisomes (Chen, 1998; Corpas & Barroso, 2018; Gálvez, Lancien & Hodges, 1999). Approximately 80–95% of the total ICDH activity in leaf tissues depends on NADP+-ICDH (Chen & Gadal, 1990). By supplying the cytosol with 2-OxG as a primary acceptor for NH4+ assimilation, ICDH is hypothesized to play a key role in the synthesis and export of amino acids, and to be a link between C and N metabolism (Fieuw et al., 1995).

Besides 2-OxG production, NADP+-ICDH is believed to be involved in supplying the cytosol with NADPH (Podgórska et al., 2013; Hodges, 2002) which is required for the reduction of NO3− or for many other biosynthesis pathways in cytosol and plastids (Kruger & Von Schaewen, 2003). This role is supported by the oxidative pentose phosphate pathway (OPPP). Two OPPP enzymes, i.e., glucose-6-phosphate dehydrogenase (G6PDH, EC 1.1.1.49) and 6-phosphogluconate dehydrogenase (6PGDH, EC 1.1.1.44), were found in the cytosol, plastids, and peroxisomes (Corpas & Barroso, 2018). G6PDH, a rate-limiting enzyme of the OPPP, oxidizes glucose-6-phosphate into 6-phosphoglucono- δ-lactone and reduces NADP+ to NADPH, whereas 6-PGDH oxidizes 6-phosphoglucono- δ-lactone into ribulose-6-phosphate and produces a second molecule of NADPH (Kruger & von Schaewen 2003; Singh & Srivastava, 2014). In higher plants, the major form of G6PDH is a cytosolic isoform representing 80–95% of total cellular activity, with the remaining 5–20% of the total activity being performed by the chloroplastic or plastidic isoform (Cardi et al., 2013).

Glutamate is a central intermediate of nitrogen metabolism (Forde & Lea, 2007), and acts as a substrate for the synthesis of amino acids (arginine, proline and glutamine), nucleotides, chlorophyll and glutathione (Martinez-Andújar et al., 2013). There is strong evidence that salt stress increases the activity of the enzymes involved in the Glu metabolism, including the aminotransaminases engaged in the production of all protein amino acids, except proline. Glu is a major amino group donor for the alanine aminotransferase (AlaAT, EC 2.6.1.2) and AspAT functions (Jha & Dubey, 2004). Reactions catalyzed by aminotransferases are reversible; alanine and aspartate may also participate in the replenishment of the Glu pool (Forde & Lea, 2007).

Agricultural productivity is severely affected by soil salinity across wide areas of land. According to United Nations Food and Agriculture Organization statistics FAO (http://faostat.fao.org), one of the most widely-cultivated vegetables is the cucumber. In Poland alone, cucumber and gherkin cultivation in 2014 encompassed 16,552 ha of land with total production being 538,057 tones; this value was the highest since 1963. Despite this importance, cucumber plants are characterized by low resistance to various stress conditions especially salinization.

It is important that plants can maintain a correct N:C ratio, and to achieve this, various biochemical processes have developed. These processes enable the plant to adjust its metabolism and accommodate environmental stress conditions (Coruzzi & Zhou, 2001).

The aim of the present study was to assess the ability of cucumber plants to acclimatize to NaCl stress by evaluating the changes of nitrate and carbon metabolism associated with this process. To this end, the study examines a range of parameters in cucumber leaves taken from stressed plants which had been either acclimated or non-acclimated to salinity stress: NADH/NAD+-GDH, AspAT, AlaAT and NADP+-ICDH, G6PDH, 6GPDH activity, and proline, Glc and Suc level were measured 24 h and 72 h after being subjected to moderate and severe NaCl stress in the two groups of plants.

Materials and Methods

Materials

Cucumber (Cucumis sativus L.) cv. “Cezar” plants were grown in a growth chamber at a temperature of 23 °C with 16 h light/8 h dark photoperiod with 350 µE m−2s−1 light intensity and 60–70% relative humidity. Five-week-old plants with four fully-expanded leaves were used. Two groups of cucumber plants were used in the study: one group that had been acclimated to salt stress (AP) by fourfold treatment with 20 mM NaCl at seven-day intervals, and another group that had not (NAP). Finally, all groups were stressed with 100 mM NaCl (moderate stress) (NAP-100, AP-100) and 150 mM NaCl (severe stress) (NAP-150, AP-150). The activities of NADH/NAD+-GDH, AspAT, AlaAT and NADP+-ICDH G6PDH and 6GPDH, and the levels of proline, Glc and Suc were examined 24 h and 72 h after moderate and severe stress application.

Methods

Enzyme extract preparation

Fresh leaf tissue (0.5 g) was homogenized in a mortar at 4 °C in 2.5 cm3 of medium: 50 mM Tris-HCl buffer (pH 7.6) containing 1 mM MgCl2, 1 mM EDTA, 1 mM DTT, 0.5% PVP (polyvinylpyrrolidone) and 10 mM β-mercaptoethanol. The homogenate was centrifuged at 20,000 g for 20 min at 4 °C. The supernatant was used to determine NADH/NAD+-GDH, AspAT, AlaAT and NADP+-ICDH, G6PDH and 6GPDH level, as well as protein content.

Glutamate dehydrogenase enzyme assay

GDH activity was assayed spectrophotometrically at 30 °C by monitoring the oxidation of NADH (aminating GDH activity, NADH-GDH) or reduction of NAD (deaminating GDH activity, NAD+-GDH) (ε = 6.22 mM−1 cm−1) at 340 nm according to Groat & Vance (1981). For NADH-GDH activity, the reaction mixture (2 cm3) consisted of 0.1 M Tris-HCl buffer, pH 8.0, enzyme extract, 11 mM 2-oxoglutaric acid, 0.1 M NH4Cl, and 0.2 mM NADH. For NAD+-GDH activity the reaction mixture (2 cm3) consisted of 0.1 M Tris-HCl buffer, pH 8.8, enzyme extract, 80 mM L-glutamic acid, and 0.7 mM NAD+. The enzyme activity was expressed in units, each representing the amount of enzyme catalyzing the oxidation/reduction of 1 nmol NADH/NAD per minute and expressed in U mg−1 protein.

Aminotransferase enzyme assay

AlaAT and AspAT activities were measured spectophotometrically according to De Sousa & Sodek (2003). AlaAT activity was assayed in the alanine-pyruvate direction by coupling the reaction with NADH oxidation by lactate dehydrogenase. The reaction mixture (2 cm3) consisted of 0.1 M Tris-HCl buffer, pH 7.5, 0.5 M L-alanine, 15 mM 2-oxoglutarate, 0.18 mM NADH, five units of lactate dehydrogenase and enzyme extract. AspAT activity was assayed in the aspartate-oxaloacetate direction by coupling the reaction with NADH oxidation by malate dehydrogenase. The reaction mixture (2 cm3) consisted of 0.1 M Tris-HCl buffer pH 7.8, 5 mM EDTA, 0.2 M L-aspartate, 12 mM 2-oxoglutarate, 0.18 mM NADH, five units of malate dehydrogenase and enzyme extract. AlaAT and AspAT activities were calculated using the absorption coefficient for NADH (ε = 6.22 mM−1 cm−1) and expressed in units, each representing the amount of enzyme catalyzing the formation of 1 nmol of product per minute and expressed in U mg−1 protein.

NADP+-dependent isocitrate dehydrogenase enzyme assay

NADP+-ICHD activity was measured spectophotometrically at 25 °C by monitoring the reduction of NADP (ε = 6.22 mM−1 cm−1) at 340 nm according to Canino et al. (1996). The reaction medium (2 cm3) contained the following: 0.05 M Tris-HCl, pH 8.0, 0.6 mM NADP+, 1mM MgCl2, 9.6 mM isocitrate as a starter of reaction. The enzyme activity was expressed in units, each representing the amount of enzyme catalyzing the reduction of 1 nmol NADP+ per minute and expressed in U mg−1 protein.

Glucose-6-phosphate and 6-phosphogluconate dehydrogenase enzyme assay

Dehydrogenase activity was assayed spectrophotometrically at 30 °C by monitoring the reduction of NADP+ (ε = 6.22 mM−1 cm−1) at 340 nm according to Sgherri et al. (2002). For G6PDH activity, the reaction mixture (2 cm3) consisted of 0.05 M Tris-HCl buffer pH 7.7, enzyme extract, 10 mM MgCl2; 0.25 mM NADP+ and 2 mM G6P as a starter of reaction. For 6PGDH activity the reaction mixture (2 cm3) consisted of 0.05 M Tris-HCl buffer pH 7.7, enzyme extract, 10 mM MgCl2; 0.25 mM NADP+ and 2 mM 6PG as a reaction starter. The enzyme activity was expressed in units, each representing the amount of enzyme catalyzing the reduction of 1 nmol NADP+ per minute and expressed in U mg−1 protein.

Sugar determinations

Glc and Suc contents were determined in the extracts obtained from leaf samples (0.5 g FW) after triple 80% ethanol extraction. The ethanolic extracts were evaporated to dryness at 50 °C and the residue was resolubilized with distilled water. The concentrations of both sugars were assayed using commercial enzymatic test (Boehring Mannheim) according to the manufacturer’s instructions and expressed in µg per mg protein determined in the extract used for enzyme activity analyses.

Proline determination

Free proline content was determined using the ninhydrin method (Bates, Waldren & Teare, 1973). The proline concentration was estimated in reference to a standard curve for L-proline and expressed in micromoles per milligram protein.

Protein determinations

Protein content was measured according to Bradford (1976) using BSA as a standard.

Statistical analysis

The significance of differences between mean values was determined by the nonparametric Mann–Whitney Rank Sum Test using Statistica 13 software (StatSoft, Tulsa, OK, USA). Differences at P < 0.05 were considered significant. Data are given as mean values ± standard deviation. Each data point is the mean of four independent experiments (n = 4) and one plant per treatment was analyzed in each experiment.

Result

Activity of aminating glutamate dehydrogenase (NADH-GDH)

The results showed that acclimation did not significantly influence NADH-GDH activity (Fig. 1A). Moreover, the activity demonstrated in the non-acclimated and acclimated plants changed to a similar degree under both NaCl stress levels. These changes were observed mainly 72 h after moderate and severe stress application. NADH-GDH activity increased by about 200% in the NAP-100 and NAP-150 groups compared to NAP, and by 208% (AP-100) and 348% (AP-150) compared to AP.

Figure 1 Changes in aminating glutamate dehydrogenase (NADH-GDH) (A) and deaminating glutamate dehydrogenase (NAD+ −GDH) (B) activities in cucumber leaves in plants treated with 100 and 150 mM NaCl: non-acclimated (NAP) and acclimated (AP) to salinization.

Bars represent SD of means, n = 4. * indicate a significant difference between NAP and AP; or NAP, NAP-100 and NAP-150; or AP, AP-100 and AP-150.

Activity of deaminating glutamate dehydrogenase (NAD-GDH)

Contrary to NADH-GDH activity, the acclimation process influenced the deaminating activity of GDH (Fig. 1B). GDH activity in the non-stressed plants was significantly lower in the acclimated then in the non-acclimated ones after both 24 and 72 h. After stress application, NAD+-GDH activity increased to a greater degree in the acclimated plants: the activity was 52% higher for AP-150 than AP after 24 h, and 137% higher after 72 h.

Activity of aminotransferases

In the non-stressed variants, no differences were found between the acclimated and non-acclimated plants for either aminotransferase (Figs. 2A and 2B). However, the difference appeared after stress application. In the non-acclimated plants, AlaAT activity decreased by about 25% after 24 h following moderate salt stress (NAP-100) compared to NAP (Fig. 2A); however, no further significant change in enzyme activity was observed for NAP-100 or NAP-150 after prolonged exposure (72 h). However, differences in AlaAT activity were observed in the acclimated plants: Enzyme activity was found to be about 70–80% greater in AP-100 and AP-150 than in AP at 72 h after exposure.

Figure 2 Changes in alanine aminotransferase (AlaAT) (A) and asparate aminotransferase (AspAT) (B) activities in cucumber leaves in plants treated with 100 and 150 mM NaCl: non-acclimated (NAP) and acclimated (AP) to salinization.

Bars represent SD of means, n = 4. * indicate a significant difference between NAP and AP; or NAP, NAP-100 and NAP-150; or AP, AP-100 and AP-150.

No change in AspAT activity was observed between the AP and NAP groups at 24 h (Fig. 2B). In contrast, after 72 h, while only moderate stress (NAP-100) increased AspAT activity in the non-acclimated plants (23% higher than NAP), both types of stress (AP-100 and AP-150) increased AspAT activity in the acclimated plants (around 70% higher than AP).

Activity of NADP+-dependent isocitrate, glucose-6-phosphate and 6-phosphogluconate dehydrogenases

Acclimation decreased NADP+-ICDH activity by about 40%. After 24 h, NADP+-ICDH activity increased by about 63% (AP-100) and 70% (AP-150) compared to AP, but no such increase was observed in non-acclimated plants. However, after 72 h, NADP +-ICDH activity was higher in NAP-100 (46%) and NAP-150 (122%) compared to NAP, and higher in AP-100 (27%) and in AP-150 (80%) than in AP (Fig. 3C).

Figure 3 Changes in 6-phosphogluconate dehydrogenase (6PGDH) (A), glucose-6-phosphate dehydrogenases (G6PDH) (B) and NADP+-isocytrate dehydrogenases activity (NADP+-ICDH) (C) activities in cucumber leaves in plants treated with 100 and 150 mM NaCl: non-acclimated.

Bars represent SD of means, n = 4. * indicate a significant difference between NAP and AP; or NAP, NAP-100 and NAP-150; or AP, AP-100 and AP-150.

The acclimation process influenced the activity of G6PDH (Fig. 3B). Moreover, acclimation influenced plant response to salt stress. After 24 h, increases in G6PDH activity were observed only in the non-acclimated plants: 48% (NAP-100) and 24% (NAP-150) compared with NAP. Following prolonged stress (72 h), both NAP and AP showed enhanced G6PDH activity, but these increases were greater in acclimated plants; however, while this activity was 120% higher in NAP-100 and 98% higher in NAP-150 compared to NAP, these levels were about 200% higher in AP-100 and AP-150 compared to AP. Stress application induced significant changes in 6PGDH activity in both groups of cucumber plants at 72 h (Fig. 3A), being significantly higher in each stressed group than its respective non-stressed group. In addition, in acclimated plants, this activity was higher than in non-acclimated ones: 6PGDH activity was about 30% higher in NAP-100 than NAP but about 70% higher in AP-100 and AP-150 than in AP.

Glucose and sucrose concentrations

The process of acclimation significantly influenced the constitutive Glc concentration in the studied variants, NAP and AP (Fig. 4A). Glc level was more than two to four times higher in the AP group than the NAP group. Following salt stress, Glc concentrations increased significantly after 24 h and were about three times higher in NAP-100 and NAP-150 than in NAP. Two days later (72 h), these values increased to twelve times higher in NAP-100 and NAP-150 compared to NAP. Similarly, in the plants acclimated to stress, the AP-100 and AP-150 Glc concentrations increased by 50% and 71%, respectively, in comparison to AP after 24 h; these levels further increased to 85% and 51% for AP-100 and AP-150, respectively, compared to AP after 72 h.

Figure 4 Changes in glucose (A), sucrose (B) and proline (C) concentrations in cucumber leaves in plants treated with 100 and 150 mM NaCl: non-acclimated (NAP) and acclimated (AP) to salinization.

Bars represent SD of means, n = 4. * indicate a significant difference between NAP and AP; or NAP, NAP-100 and NAP-150; or AP, AP-100 and AP-150.

Similarly to Glc, sucrose concentration was found to be two- to threefold higher in AP than NAP. Following stress, greater changes were observed after 72 h. Interestingly, in NAP-100, salt stress caused a progressive increase in Suc concentration, rising from almost twice higher than NAP after 24 h to twenty-times higher after 72 h. In response to the severe stress (NAP-150) Suc concentration increased significantly, being eighteen-fold higher in comparison to NAP after 72 h (Fig. 4B). A high level of Suc accumulation was observed in the acclimated plants, but only after 72 h; however, after this time, Suc concentration increased in all exposed groups, being four-fold higher in AP-100 and twelve-fold higher in AP-150 compared to AP values.

Proline content

The acclimation process significantly influenced proline content (Fig. 4C). Constitutive Pro level was 20–35% lower in AP than in NAP; however, this value increased during the experiment, reaching its highest concentration 72 h after stress treatment. After 24 h, both acclimated and non-acclimated plants demonstrated a significant increase in free proline level after severe stress treatment compared to respective controls (63% for NAP-150 and 49% for AP-150). After 72 h, the Pro level was 395% higher in NAP-100 and 496% higher in NAP-150 compared to NAP, and 308% higher in AP-100 and 395% higher in AP-150 compared to AP.

Discussion

Salt stress is associated with the accumulation of compatible solutes such as carbohydrates, and this is generally considered to be an adaptive response to salinization (Roitsch, 1999). A high level of carbohydrates inhibits the expression of genes coding for photosynthesis enzymes, including those of the Calvin cycle (Couee et al., 2006).

In our study, the accumulation of soluble carbohydrates (Glc and Suc) was observed after stress (Figs. 4A, 4B). It should be noted that while both carbohydrates were initially present in higher amounts in the acclimated plants, a smaller increase in sugar levels was observed in the acclimated plants than the non-acclimated ones following exposure, and that the two carbohydrates accumulated at different rates: Glc concentration increased early (24 h) in all stressed groups, whereas Suc accumulation varied according to acclimation, stress intensity and time of exposure. It should be noted that the highest Suc level was observed with the lowest Glc level only in AP-150 after 72 h. This might indicate that in the acclimated plants, severe stress led to reverse regulation of these carbohydrates and enhanced direct utilization of Suc in adaptive processes. Previous studies indicate that, contrary to Glc, Suc is an effective osmoprotectant (Singh et al., 2015; Sami et al., 2016).

Acclimation decreased NADP+-ICDH activity, which suggests that this enzyme is highly susceptible to salinity stress. However, in both groups of plants (NAP and AP) the NADP+-ICDH activity was found to be enhanced during the later phase (72 h) of the experiment, but was lower in AP than in NAP. This reduction in NADP+-ICDH activity associated with acclimation might result from salt-triggered inhibition of aconitase, which supplies a substrate for isocitrate dehydrogenase by salt-induced generation of ROS, especially H2O2 (Zhang et al., 2009). The cytosolic isoform of NADP+-ICDH is an important NADP+ reducing enzyme, and together with OPPP enzymes, replenishes the pool of NADPH. Besides fuelling NADPH-dependent enzymes e.g., in proline synthesis, the NADPH produced by mitochondrial and chloroplastic ICDH activity might be important for glutathione regeneration used in inter alia the ascorbate-glutathione cycle and glutathione peroxidase system (Gálvez & Gadal, 1995; Corpas & Barroso, 2014), both of which are involved in ROS scavenging in mitochondria, chloroplasts and cytosol (Hodges et al., 2003). Liu et al. (2010) showed that, in transgenic Arabidopsis thaliana, over-expression of cytosolic NADP+-ICDH (ZmICDH) induced by drought and salt stress boosted salt tolerance.

Accumulation of soluble sugars under salt stress is known to enhance free Pro content (Hellmann et al., 2000). Proline accumulation ameliorates the damaging effect of salt stress by functioning as an osmoprotectant in a similar way to Suc. However, it remains undecided whether its protective influence is due to the direct scavenging of reactive oxygen species (ROS) (Hayat et al., 2012; Signorelli et al., 2016). Pro has been found to contribute to the scavenging of .OH by a Pro-Pro cycle without consumption of Pro, and not to quench singlet oxygen (Signorelli et al., 2014). In addition, proline accumulation increases under saline conditions but is not a direct scavenger of peroxinitrite, superoxide, nitrogen oxide or dioxide (Signorelli et al., 2013; Signorelli et al., 2016). However, high Pro accumulation might have a negative influence on a cell, possibly leading to protein denaturation (Hayat et al., 2012) and changes in mitochondria and chloroplast ultrastructure, and may also initiate the early steps of programmed cell death (Deuschle et al., 2004). Although proline concentration was found to increase in both stressed groups, Pro concentration was lower in the acclimated plants. Zhang et al. (2009) report that disrupted functioning of mitochondria influences carbon and nitrogen metabolism and cellular biosynthesis. This disruption may decrease TCA cycle activity and disturb ATP production. Salt stress leads i.a to changes in the profiles of organic acids which are mainly produced in the TCA cycle (López-Bucio et al., 2000). Moreover, as malate stimulates nitrate uptake by the root (Touraine, Muller & Grignon, 1992), demand for malic acid is enhanced when nitrogen is limited by salinity. As malate is synthesized via malate dehydrogenase from oxaloacetate and NADH in the presence of NAD+, it is another process which might influence the activity of the TCA cycle (Van Dongen et al., 2011). Generally, reduced TCA cycle activity activates a fermentation processes which leads, among others, to an accumulation of alanine via a reversible reaction converting pyruvate and glutamate to alanine and 2-OxG; the process is catalyzed by AlaAT. As the formed 2-OxG may enter into the TCA cycle, AlaAT activity facilitates the production of ATP (Van Dongen et al., 2011). However, alanine synthesis is not the sole cause of AlaAT induction (De Sousa & Sodek, 2003): As reversible transamination reactions are controlled by substrate and product level, it is possible that a rapid increase in pyruvate or depletion of 2-OxG could also be responsible (De Sousa & Sodek, 2003). Although AP and NAP demonstrated similar AspAT and AlaAT levels, changes were observed under stress situations (Figs. 2A, 2B): In acclimated plants, similar increases in AspAT and AlaAT activity were observed in AP-100 and AP-150 after 72 h, while in non-acclimated plants, AlaAT activity decreased in NAP-100 and NAP-150 after 24 h, and AspAT increased only in NAP-100 after 24 h.

In the acclimated plants, the glutamate-producing AlaAT and AspAT processes seem to play an important role in the replenishment of the glutamate pool. Glutamine is not only engaged in the assimilation and release of ammonia, it is also a substrate in proline synthesis. Our results indicated that increases in AlaAT, AspAT and aminating GDH activities correlated with the growth of free proline level. In many plant species, Pro accumulation associated with salt stress has been correlated with stress tolerance, and its concentration is generally higher in salt-tolerant plants than in salt-sensitive ones (Szabados & Savoure, 2010). Free proline level has previously been found to correlate with high vigour in acclimated suspension cell cultures subjected to salinization (Naliwajski & Skłodowska, 2014).

Our present data indicates that free proline level in leaf tissues increased in response to salt stress, especially in the late phase of the experiment. The presence of fewer salt-induced injury symptoms in the acclimated plants, such as the presence of chlorotic spots on the leaf surface, yellowing starting at the leaf tips and margins and working back to the vein and wilting (Figs. 5 & 6), might be associated with an elevated concentration of sucrose, which is a better osmoprotectant than glucose (Rejšková et al., 2007; Keunen et al., 2013; Singh et al., 2015). Degl’Innocenti et al. (2009) report that Hordeum maritimum showed a lower reduction of growth under saline conditions compared to Hordeum vulgare, indicating substantial salt tolerance.

Figure 5 Effect of salt stress treatment on the morphology of non-acclimated cucumber plants.

NAP, non-acclimated plant; NAP-100, non-acclimated plant stressed with 100 mM NaCl; NAP-150, non-acclimated plant stressed with 150 mM NaCl. (A) NAP—24h; (B) NAP-100—24h; (C) NAP-150—24h; (D) NAP—72h; (E) NAP-100—72h; (F) NAP-150—72h.

Figure 6 Effect of salt stress treatment on the morphology of acclimated cucumber plants.

AP, acclimated plant; AP-100, acclimated plant stressed with 100 mM NaCl; AP-150, non-acclimated plant stressed with 150 mM NaCl. (A) AP—24h; (B) AP-100—24h; (C) AP-150—24h; (D) AP—72h; (E) NAP-100—72h; (F) AP-150—72h.

The production and accumulation of organic solutes is one of the important physiological responses demonstrated by plants to salinity; however, it is an energy-consuming process which decreases growth. Pérez-López et al. (2009) report that the observed decrease in plant growth resulted not only from the toxic effect of ions, but also from poor water relations under salt stress. The lower level of Pro observed in the acclimated plants compared to non-acclimated ones following salt stress might be associated with the use of glutamate, a substrate for proline synthesis, to other metabolic processes i.a. synthesis of arginine (Arg) which acts as a major nitrogen storage compound in higher plants. Moreover, arginine, together with ornithine, may also act as a precursor of polyamines, which play an important role in plant stress tolerance (Forde & Lea, 2007). The fact that the level of Pro was lower in stressed AP than in stressed NAP may result from its possible metabolism to prolinebetaine and hydroxyprolinebetaine in AP: the two compounds are more potent osmoprotectants than proline. Research on related species with different capacities for prolinebetaine and hydroxyprolinebetaine accumulation strongly indicates that the free proline pool size is inversely related to the total pool of the two compounds (Snight, 1999). The lower concentration of proline observed in the acclimated plants than in the non-acclimated ones cannot be connected with its enzymatic degradation or a decrease in the activity of proline synthesis enzymes (Szabados & Savoure, 2010). The changes observed in AlaAT activity in the non-acclimated cucumber plants show that the TCA cycle might be disturbed, and that under stress, NADH-GDH and AspAT are mainly engaged in the synthesis of glutamate. Kumar, Shah & Dubey (2000) report that aminating GDH activity rose with increasing salt stress in salt-tolerant rice (Oryza sativa L.) cultivars. This is in agreement with our present results (Fig. 1A), which indicate that NADH-GDH activity grew in both examined groups of plants in response to salt stress.

The enhanced activity of NADH-GDH observed in both acclimated and non-acclimated plants 72 h after NaCl application suggests not only a high demand for glutamate but also the presence of a significant amount of ammonium in the cells. Under stressful conditions, including salinity, NADH-GDH can play an important role in the detoxification of ammonium originating from the degradation of organic nitrogen compounds in response to stress, as well as in the replenishment of the pool of glutamate; this acts as a substrate in the synthesis of proline, which significantly accumulates in salt-stressed plants (Frechilla et al., 2002; Lasa et al., 2002). Similarly to NADH-GDH, deaminating GDH activity was also enhanced in the acclimated cucumber leaves under severe salt stress conditions. Moreover, the simultaneous increase in NAD+-GDH activity, which catalyzes the deamination of glutamate, could also be a source of ammonium ions, as well as of 2-OxG in these organs. Bechtold, Pahlich & Lea (1998) indicate that the GDH-mediated deamination of glutamate to ammonium and 2-OxG to be very important, particularly in plants under stress conditions and during senescence.

The lack of changes in NAD+-GDH activity observed in the non-acclimated plants in response to moderate and severe stress is in agreement with the results described by Debouba et al. (2006), who report that deaminating GDH activity greatly decreased with increasing NaCl concentration in tomato leaves and roots. Our analysis of deaminating GDH activity in the stressed acclimated and non-acclimated plants revealed differences in the regulation of cellular NH4+ concentration; however, increased NAD+-GDH activity might also result from the need to replenish the limited pool of carbon skeletons, which is directly connected with the difference in C/N status. Jha & Dubey (2004) suggest that GDH isoenzymes might be differentially induced depending on stressful conditions or according to the availability of NH4+. Therefore, in the leaves exposed to salinity, some of the GDH isoforms can be induced in the direction of ammonium assimilation and others in the direction of deamination. At the end of our experiment, both NADH- and NAD+-GDH activities were substantially increased in the leaves of cucumber plants exposed to salinity. This is consistent with the parallel enhancement of amination and deamination of GDH observed in plants treated with heavy metals (Kwinta & Koźlik, 2006; Gajewska & Skłodowska, 2009).

It is probable that these changes in GDH activity take place in NaCl stressed plants. Acclimation did not influence G6PDH and 6PGDH activities, but the levels of both enzymes increased in both groups under stress conditions. However, in the acclimated plants, significant increases in both enzymes were observed only in the later phase of experiment (72 h), whereas in the non-acclimated plants, G6PDH activity was increased at 24 h while 6PGDH changed after another two days (72 h) (Figs. 3A, 3B). These findings indicate a demand for the reduced form of NADPH in the stressed non-acclimated plants from the beginning of the experiment, and that this demand continued to grow, to satisfy a demand for this nucleotide later on. Moreover, under stress conditions, higher activities of G6PDH and 6PGDH were observed in the acclimated cucumber plants than in the non-acclimated plants; this may indicate that the reduced pool of G6PDH was replenished more quickly in the AP plants. Upon exposure to salt, the cytosolic G6PDH activity and transcript level may increase in order to support the syntheses of cofactors or intermediates involved in the tolerance mechanisms (Nemoto & Sasakuma, 2000; Dal Santo et al., 2012). G6PDH activity has also been found to be enhanced by an increase in ammonium assimilation (Bowsher et al., 1992; Esposito et al., 2001). G6PDH activity was detected in both cytosol and plastids but the changes in G6PDH activity observed in the present study were limited to the cytosol: to obtain the supernatant, it was necessary to use reduced dithiothreitol, which inactivates the plastid isoform but not the cytosolic isoform (Asai et al., 2011). G6PDH controls carbon flow in the pentose phosphate pathway and produces reducing equivalents in the form of NADPH which are used i.a. in the antioxidant pathways (Corpas & Barroso, 2018).

The oxidative damage induced by salt stress increased NADPH synthesis by OPPP; it can be hypothesized that the need to maintain a steady-state level of H2O2 in cells during stress would make hydrogen peroxide function as a signal enhancing G6PDH expression and activity (Valderrama et al., 2006). Under our experimental conditions, the activities of G6PDH and 6PGDH were significantly enhanced in the leaves of both examined groups of plants. A similar response was reported in other plant species: In two-week-old wheat (Triticum aestium L. cv. Chinese Spring) seedlings, the level of G6PDH transcripts rapidly increased within two h of NaCl treatment and reached peak expression after 12 h (Nemoto & Sasakuma, 2000). During prolonged NaCl stress, this expression fell below baseline level, indicating that G6PDH was involved in the initial responses of salt-stressed plants. However, salinity stress can cause a contrary response in other plant species: Zhang et al. (2013) found G6PDH activity to be downregulated in salt-stressed suspension cultures of rice cells. Moreover, both NADP-dehydrogenase activities were diminished in tomato root (Manai, Gouia & Corpas, 2014).

The changes in Glc and Suc levels as well as NADP+-ICDH, AlaAT and AspAT activities might indicate that after salinity stress, both groups of cucumber activated the responses. Under NaCl stress, the cells began processes that yielded intermediates for the anaplerotic reaction of the TCA cycle. Production of oxaloacetate from pyruvate catalyzed by AlaAT, transamination of aspartate to oxaloacetate by AspAT and production of 2-OxG from glutamate by NAD+-GDH, the most important physiological anaplerotic reactions producing intermediates for the TCA cycle, additionally yield the reduced form of NADH (Van Dongen et al., 2011).

The observed accumulation of Glc and Suc levels did not result from more effective photosynthesis due to the influence of osmotic stress and the presence of the toxic Na+ ion derived from the NaCl stress (Rejšková et al., 2007). Moreover, salinity stress would also decrease photosynthetic activity because it would reduce the uptake of nitrogen compounds, and CO2 assimilation is related to the nitrogen status of the leaf (Coruzzi & Zhou, 2001; Cruz et al., 2003; Li et al., 2013). The observed increases in the Glc and Suc levels were another factor which caused feedback inhibition of carbon assimilation (Foyer, 1988).

Taken together, the accumulation of carbohydrates under stress conditions seems to result from changes in their usual metabolism and/or their remobilization from storage pools. Our findings suggest that the acclimation of cucumber plants to NaCl stress has a positive function because it may prevent the destabilization of the cell respiration metabolism and balance the anaplerotic reaction which feeds the TCA cycle.

Supplemental Information

Supplemental Information 1 6-phosphogluconate dehydrogenase results

Click here for additional data file.

Supplemental Information 2 Alanine aminotransferase results

Click here for additional data file.

Supplemental Information 3 Asparate aminotransferase results

Click here for additional data file.

Supplemental Information 4 Glucose-6-phosphate dehydrogenases results

Click here for additional data file.

Supplemental Information 5 Glucose results

Click here for additional data file.

Supplemental Information 6 Deaminating glutamate dehydrogenase results

Click here for additional data file.

Supplemental Information 7 Aminating glutamate dehydrogenase results

Click here for additional data file.

Supplemental Information 8 NADP+-isocytrate dehydrogenases results

Click here for additional data file.

Supplemental Information 9 Proline results

Click here for additional data file.

Supplemental Information 10 Sucrose results

Click here for additional data file.

Abbreviations:

AlaAT alanine aminotransferase

AspAT aspartate aminotransferase

G6PDH glucose-6-phosphate dehydrogenase

GDH glutamate dehydrogenase

Glc glucose

GS/GOGAT glutamine synthetase/glutamate synthase cycle

ICDH isocitrate dehydrogenase

OPPP oxidative pentose phosphate pathway

2-OxG oxoglutarate

6PGDH 6-phosphogluconate dehydrogenase

Pro proline

ROS reactive oxygen species

Suc sucrose

TCA tricarboxylic acid

Additional Information and Declarations

Competing Interests

Author Contributions

Data Availability

The authors declare there are no competing interests.

Marcin Robert Naliwajski conceived and designed the experiments, performed the experiments, analyzed the data, contributed reagents/materials/analysis tools, prepared figures and/or tables, authored or reviewed drafts of the paper, approved the final draft.

Maria Skłodowska analyzed the data, contributed reagents/materials/analysis tools, authored or reviewed drafts of the paper.

The following information was supplied regarding data availability:

The raw data are provided in the Supplemental Files.

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
