# Peer review of "The relationship between carbon and nitrogen metabolism in cucumber leaves acclimated to salt stress"

_PeerJ, doi:10.7717/peerj.6043_

## Round 0.1 · original submission · Major Revisions

The reviewers have made some good comments that should be followed to improve the readability of the manuscript. The grammar needs improvement in many places; some of those issues are simple spelling and punctuation errors. I would also suggest updating the literature cited section with more recent research and focus the discussion section so that it is not so long.

I agree with one of the reviewers that ANOVA/LSD may not be the most appropriate for comparing the small number of observations and that additional justification (or re-analysis using a non-parametric test) is needed.

Reviewer 1 ·

Basic reporting

See general comments

Experimental design

See general comments

Validity of the findings

See general comments

Additional comments

The manuscript by Naliwajski and Skłodowska entitled “Relationship between carbon and nitrogen metabolism in cucumber leaves acclimated to salt stress” uses different biochemical approaches to study how cucumber plants response to salinity stress.
The manuscript is well designed and the biochemical methods are appropriated. I have only some minor suggestions. They are as follows:
1. Introduction section. The presence of NADP-ICDH, G6PDH and 6PGDH has been also described into plant peroxisomes. See Corpas and Barroso (2018) Peroxisomal plant metabolism - an update on nitric oxide, Ca2+ and the NADPH recycling network. J Cell Sci. 131(2). pii: jcs202978 and cited references. This could be mentioned considering the oxidative metabolism of this subcellular organelle which are involved in the mechanism of oxidative stress.
2. Describe with more details the used methods because it will be very useful to the potential readers. For example include concentration of substrate, used wavelength in the spectrophotometric assays, etc.
3. Result section Describe the panels of the figures in same order they appear in the figures
4. On the Y axis, write period instead comma in the Figs. 2A, 2B, 3C and 4.
5. Figs 5 and 6 show the phenotype of cucumber plants they should be moved to first and second figures with an appropriated description.
6. In the different figures showing enzymatic activities, on the Y axis, write the specific activity with the products or substrate used to its assay. For example write “nmol NADPH min-1 mg-1 protein” for the different NADP-dehydrogenases. This will allow to compare the present data with that already reported in other plant species.
7. Authors could explain why cucumber plants have been used in this study. Maybe because cucumber is a model to study salinity or for its agronomical relevance…
8. L336 Write ascorbate-glutathione cycle instead “Foyer-Halliwell-Asada cycle”.
9. L343-344. Very recently it has been described that Pro is not a ROS scavenger. See Signorelli et al. (2016) In vivo and in vitro approaches demonstrate proline is not directly involved in the protection against superoxide, nitric oxide, nitrogen dioxide and peroxynitrite. Functional Plant Biology 43(9) 870-879. Therefore, this statement must be deleted.
L315-316. On the other hand, similar consideration must be taken about sucrose which “is an effective ROS scavenger”. It is possible to speculate but this should be corroborated as experimental level. Therefore, this should be mentioned.
10. In the figures, the label to identify each column should be bigger it is difficult to distinguish. Moreover, it is suggested to change the lines inside of the columns, maybe color could be an alternative.
11. L72. Write matrix instead “marix”
12. Add a list of abbreviation

Reviewer 2 ·

Basic reporting

The English language should be improved to ensure that an international audience can clearly understand your text. Authors did not pay any attention to punctuation, grammar mistakes and spelling. I recommend to check the text by native English speaker.

Literature in many cases is quite old. I recommend to check and find more latest papers to base on.

The references mentioned below does not exist in the text:
'Bonnefont-Rousslot D. 2002. Glucose and reactive oxygen species. Current Opinion in Clinical Nutrition & Metabolic Care 5(5):561-568 zastąpić Krugerem'

'Bowsher CG, Hucklesby DP, Emes MJ. 1989. Nitrite reduction and carbohydrate metabolism in plastids purified from roots of Pisum sativum L. Planta 177(3):359-366.'

'Crawford NM, Arst HNJ. 1993. The molecular genetics of nitrate assimilation in fungi and plants. Annual Review of Genetics 27:115-146.'

'70. Mena-Petite A. 2009. The oxidative stress caused by salinity in two barley cultivars is 71. mitigated by elevated CO2. Physiologia Plantarum 135:29-42' - this reference is divided into two lines.

'Singh S, Srivastava PK. 2014. Purification and characterization of glucose-6-phosphate dehydrogenase from pigeon pea (Cajanus cajan) seeds. Advances in Enzyme Research 2(4):134-149'

Line 318 there is 'van Dongen' instread of 'Van Dongen'
Reference No. 23 Dal Santo is in reference, but in the text is Del Santo, please correct.
Reference No. 27 deSousa - between de and Sousa should be a space.
Reference No. 73 In the text there is Podgórska, but in reference is Podgorska, please correct.

Experimental design

There is no clear aim of the study. Authors emphasized that they did some experiments, but they did not mentioned what is the point of their research.



Why authors described time after stress treatment as 24th or 72nd. It's unnecessary. I recommend write 24 or 72 h after treatment as they did in figures description. I would try to unify the background of pictures.

Validity of the findings

Authors did grate job performing analysis, however I am not convinced why they used ANOVA and LSD test for such small group of repeats. Four repeats are too small to use parametric tests. Besides, unlike the Bonferroni, Tukey, Dunnett and Holm methods, Fisher's LSD does not correct for multiple comparisons. I recommend to analyze the data via one of the nonparametric tests.

Additional comments

No comment.

Reviewer 3 ·

Basic reporting

-The ms is well written in general, although some basic proof-reading is needed as minor grammar and syntax errors can be found throughout (as early as in the abstract section - see showed high degree of... accumulation was found).
- The introduction is too long and needs some trimming. Contrarily, I would like to see some sort of information on the process of acclimation.
- Article structure is fine, although it would be nice to include a metabolic pathway including the enzymes examined herein to get a better idea of components covered.

Experimental design

- No queries in regard with experimental design. One point though: on what basis was the acclimation protocol followed? Was this based on prior literature? If not, did the authors carry out preliminary experiments?

Validity of the findings

- Data presented is robust and appears to be statistically sound. Conclusions discussed are well-stated; however, the Discussion section is too long and should be more concise.

Additional comments

- Authors quantify proline, glucose and sucrose. On that basis, they could also carry out enzymatic activity assays of proline biosynthetic enzymes (such as P5CR and P5CS) which would be interesting, while the same goes for suc-glu-related enzymes such as invertases. Would the authors care to comment in regard?
- Furthermore, I was a bit surprised to see that damage phenotypes were limited to loss of turgor. What about other salinity-related symptoms such as yellowing and/or leaf curling?

---

## Round 0.2 · Minor Revisions

The reviewers have made some additional comments to your article that I think are relevant. I hope you are willing to address these comments and revise your manuscript accordingly.

Reviewer 1 ·

Basic reporting

The authors have included my previous suggestions, although I make some additional minor recommendations. They are as follows:
- L120 Write “16,552” instead “16552”
-L182 Write isocitrate instead isicitrate
-Throughout the text write “h” instead “hours”. For example L273
-In figure 2 and check again the units for specific activity because it is missing min-1. For example, write “nmol NADH min-1 mg-1 protein” instead “nmol NADH mg-1 protein”. Check also in Fig. 4. Write period instead comma in the Y axis

Experimental design

No comment

Validity of the findings

No comment

Reviewer 3 ·

Basic reporting

As this is a revised version, any comments are shown in the General Comments to the Authors section.

Experimental design

As this is a revised version, any comments are shown in the General Comments to the Authors section.

Validity of the findings

As this is a revised version, any comments are shown in the General Comments to the Authors section.

Additional comments

I am generally satisfied with the way the authors handled my comments. However, I do not fully follow the response received in regard with my suggestion to carry out enzymatic activity assays for proline and sugar metabolism components. I can understand (and accept) the comment about sugar metabolism being currently examined as part of ongoing work. However, if I understood correctly, the authors did carry out p5CS and PDH activity assays but choose not to include these (although in the response letter they strangely mention that the results will be included in the prepared manuscript). I am confused: will they or won't they include these? If not, why? Doing so would greatly support the story being told as we would get biochemical evidence for proline regulation under the current experimental setup.

---

## Round 0.3 · accepted · Accept

Thank you for your efforts in revising the manuscript

#